# Brain-Derived 11S Regulator (PA28αβ) Promotes Proteasomal Hydrolysis of Elongated Oligoglutamine-Containing Peptides

**DOI:** 10.3390/ijms241713275

**Published:** 2023-08-26

**Authors:** Viacheslav A. Kriachkov, Natalia N. Gotmanova, Vadim N. Tashlitsky, Anna V. Bacheva

**Affiliations:** 1Department of Biochemistry and Molecular Biology, Bio21 Molecular Science and Biotechnology Institute, University of Melbourne, Melbourne, VIC 3010, Australia; vkriachkov@student.unimelb.edu.au; 2Department of Chemistry, Lomonosov Moscow State University, Leninskie Gory 1, 119991 Moscow, Russia; got.nataliia@gmail.com (N.N.G.); tashlitsky@belozersky.msu.ru (V.N.T.)

**Keywords:** huntingtin, poly-glutamine disease, proteasome, 11S regulator

## Abstract

Proteins with extended polyglutamine regions are associated with several neurodegenerative disorders, including Huntington’s disease. Intracellular proteolytic processing of these proteins is not well understood. In particular, it is unclear whether long polyglutamine fragments resulting from the proteolysis of these proteins can be potentially cleaved by the proteasome. Here, we studied the susceptibility of the glutamine-glutamine bond to proteolysis by the proteasome using oligoglutamine-containing peptides with a fluorophore/quencher pair. We found that the addition of the 11S proteasomal regulator (also known as PA28) significantly accelerated the hydrolysis of oligoglutamine-containing peptides by the 20S proteasome. Unexpectedly, a similar effect was observed for the 26S proteasome in the presence of the 11S regulator. LC/MS data revealed that the hydrolysis of our peptides with both 20S and 26S proteasomes leads to N-terminal fragments containing two or three glutamine residues and that the hydrolysis site does not change after the addition of the 11S regulator. This was confirmed by the docking experiment, which shows that the preferred hydrolysis site is located after the second/third glutamine residue. Inhibitory analysis revealed that trypsin-like specificity is mainly responsible for the proteasomal hydrolysis of the glutamine-glutamine bond. Together, our results indicate that both 20S and 26S proteasomes are capable of degrading the N-terminal part of oligoglutamine fragments, while the 11S regulator significantly accelerates the hydrolysis without changing its specificity. This data suggests that proteasome activity may be enhanced in relation to polyglutamine substrates present in neurons in the early stages of polyglutamine disorders.

## 1. Introduction

The control over the level of intracellular proteins is mediated by both the ubiquitin-proteasome system and autophagic degradation; the latter occurs only for relatively large components in the cytoplasm [1,2]. Proteasomes are a group of complex, multi-catalytic threonine endopeptidases (E.C. 3.4.25.1) and consist of a proteolytic 20S subparticle and either zero, one, or two regulators (identical or different) known as 11S (PA28), 19S (PA700) and PA200 [3]. The core proteolytic subparticle consists of four stacked rings, each containing seven proteins of two different types, alpha and beta. Three of the seven β subunits in each inner ring are catalytically active and have an identical catalytic mechanism but different substrate specificity: caspase-like (β1), trypsin-like (β2), or chymotrypsin-like (β5). The actual substrate specificity of a proteasome is even broader, which allows it to hydrolyze any protein into short peptides. The failure of protein hydrolysis by proteasomes may lead to cell apoptosis or various pathologies, including neurodegenerative disorders. Although most abnormal proteins are cleaved by the proteasome, sometimes this process is inefficient as in the case of several neurodegenerative diseases [4]. Diseases associated with polyglutamine (polyQ) repeats are genetic neurodegenerative disorders resulting from the expansion of CAG repeats in some genes, which leads to the presence of long polyQ tracts (from 6–35 to 36–306 glutamine residues) in corresponding proteins. These mutant proteins are prone to aggregation and form insoluble inclusion bodies in neurons. The analysis of these inclusions has shown the presence of ubiquitin and 26S proteasome components (20S core, 19S, and 11S regulatory particles) [5,6,7].

The accumulation of aggregates, both in the nucleus and in the cytoplasm [8,9], leads to the inefficient operation of proteolytic degradation systems in the affected neurons. It has been suggested that the proteasome degradation system, which usually degrades abnormal and misfolded proteins, fails to efficiently digest substrates with extended polyQ regions and that these polyQ tracts can inactivate proteasome complexes by clogging the catalytic chamber of the 20S core particle [10]. It has also been shown that the hydrolysis of polyQ-containing proteins and peptides occurs at random sites and at a slow rate, meaning that such proteins occupy the proteasome for a longer time [11,12], thus indirectly diminishing proteasome activity. The question of whether the proteasome can hydrolyze the polyQ sequence remains unresolved. While some studies observed proteasome inhibition by polyglutamine-containing proteins [11,12,13,14,15,16], others suggested that proteasomes are capable of hydrolyzing polyglutamine sections of any length [17,18,19,20,21]. Such a discrepancy could partly be attributed to the different study models used by researchers; however, the inability of the proteasome to degrade mutant huntingtin efficiently has been demonstrated using a wide variety of models. Each of the used models has its own drawbacks, and cell lines such as HeLa [12], fibroblasts [13], and HEK293T cannot adequately reflect what happens in neurons. In vitro models have their own limitations since the substrates used in these models are often non-ubiquitinated, and it is well established that many cellular proteins, when non-ubiquitinated or improperly ubiquitinated, do not undergo proteasome degradation. Another possible reason is that most of these studies did not investigate the proteasome subunit composition during the degradation of polyglutamine-containing proteins. It is known that the replacement of the constitutive catalytic β1, β2, and β5 subunits by the immune subunits (β1i, β2i, and β5i) changes the proteolysis rate, the proteasome specificity, and the length of the resulting fragments [22,23].

In the cell, most 20S proteasomes function in complexes with special regulatory proteins that, when attached to the main 20S subparticle, open the entrance into the proteolytic chamber and can impact the catalytic properties of the entire enzyme. One such regulator is 19S, which has ATPase activity and a ubiquitin-recognition site. Together with the 20S proteasome, it forms a complex known as the 26S proteasome, which is central to the ubiquitin-dependent degradation of proteins [2]. Another regulatory protein, 11S, otherwise known as REG protein or PA28 (a proteasome activator with an apparent subunit molecular weight of 28 kDa), represents a ring-shaped heptamer. The 11S family of regulatory proteins consists of two related complexes: cytosolic γ-interferon-inducible 11Sα/β, which is a stochastic heteroheptamer, and nuclear homoheptamer 11Sγ [1,2]. Both 11S types can increase the rate of proteasome hydrolysis for substrates without ubiquitin labels and in the absence of ATP due to binding to an α-ring of the 20S proteasome that opens the gate into the central proteolytic chamber. Hybrid 19S-20S-11S proteasome complexes can also form; in this case, 19S is responsible for proteins’ unfolding and extension into the chamber of the 20S subparticle, while the 11S regulator participates in the release of the reaction products. It has been proposed that the 11S regulator allows the proteasome to degrade substrates containing long polyglutamine sequences [19]. This assumption needs to be further verified, and the aim of our study is to rigorously investigate the effect of the 11S regulator on the proteasomal degradation of peptides containing oligoglutamine sequences.

## 2. Results

### 2.1. Substrate Characterization

We developed a new, internally quenched fluorogenic substrates to examine the proteasomal degradation of peptides with oligoglutamine fragments of various lengths, both in the presence and absence of the 11S regulator protein. We used peptide substrates of the following composition: Dabcyl-KQ5GD-EDANS, Dabcyl-KQ10GD-EDANS, and Dabcyl-KQ10PPD-EDANS. Each of them contained a widely used EDANS-Dabcyl FRET pair and an oligoglutamine fragment. The composition of Dabcyl-KQ10PPD-EDANS with two proline residues after the oligoglutamine sequence resembles the primary structure of huntingtin protein, where a polyproline tract is located right after the polyglutamine sequence [24]. FRET efficiency (*E*) for each of the substrates can be calculated using the following Equation (1):(1)$$E=\frac{1}{1+{\left(\frac{r}{R_{0}}\right)}^{6}}$$
where *r* is the distance between the donor chromophore (EDANS) and acceptor chromophore (Dabcyl), and *R*_0_ is the Förster distance of the EDANS-Dabcyl pair, which is equal to 33 Å [25]. The peptide length was calculated using Scratch Protein Predictor, a structure prediction tool [26] (Figure 1).

Both longer peptides were expected to form helical structures since there are ten consecutive glutamine residues within each of them, while the smaller peptide with five glutamines was presumed to have an insufficient length to form a stable helix. The distances (*r*) between EDANS and Dabcyl were proven to be sufficient to provide an effective energy transfer (Table 1).

To further characterize our modified peptides, we examined the change in fluorescence signal between the starting substrate solution and the totally hydrolyzed substrate. The fluorescence spectra of peptides after complete hydrolysis by subtilisin, a non-specific protease, had an emission maximum in the range of 490–510 nm, which is associated with the fluorescence of the EDANS moiety (Appendix A). The fluorescence intensity at λ_max_ increased 13.3-fold after the hydrolysis of Dabcyl-KQ5GD-EDANS, and the signal was amplified 8-fold for substrates with ten glutamine residues. Therefore, the experimental data are in good agreement with the theoretical calculations. Hence, these substrates can be used to estimate the ability of a proteasome to hydrolyze peptides carrying an oligoglutamine fragment. Some additional characteristics of these substrates can be found elsewhere [27].

### 2.2. 11S Protein Increases 26S Proteasomal Activity towards Longer Peptides

It is well known that ATP-independent 11S regulator proteins stimulate 20S peptidase activity [28,29], but the data from studies of 26S proteasome activity with 11S is contradictory [30,31]. To study the effect of the 11S regulator protein on the activities of 20S and 26S, we measured the rate of proteasomal cleavage of FRET substrates at different molar ratios between the 11S protein and the proteasome (Figure 2). Both 20S and 26S proteasomes, as well as 11S regulator, were isolated from mouse brains. The isolated 20S proteasome was in a closed-gated form since it showed strong activation of protease activity upon the addition of 0.02% SDS (Appendix A) and mostly consisted of constitutive subunits (Appendix A). We showed that isolated 11S consisted of α/β subunits using dot-blotting (Appendix A). Isolated 26S proteasomes represented a mixture of single- and double-capped 26S particles and did not contain single uncapped 20S proteasome particles (Appendix A), as it was strongly inhibited by the addition of 0.02% SDS (Appendix A). The concentrations of both 20S and 26S proteasomes in this assay were equal to approximately 14 nM.

For a short FRET substrate consisting of eight amino acid residues (Dabcyl-KQ5GD-EDANS), 20S proteasome activity increased with the growing number of 20S+11S complexes (Figure 2a). At the same time, the rate of its degradation by the 26S proteasome did not change in the presence of 11S. On the contrary, the cleavage rate of the longer substrate consisting of 14 amino acid residues (Dabcyl-KQ10PPD-EDANS) increased with the growing concentration of 11S in both 20S and 26S proteasomes (Figure 2b). To determine whether this difference between the two peptides is due to the length of the whole peptide or to the length of the oligoglutamine fragment, we studied the effect of the 11S regulator on the degradation of two other peptides lacking the oligoglutamine fragment (Figure 2c,d). Both substrates were degraded faster by the 20S+11S complex than by the 20S proteasome. We also observed a notable activation of the 26S proteasome by 11S for the longer HIV-protease substrate but not for the short substrate Suc-LLVY-AMC.

### 2.3. The Effect of 11S on the Kinetic Parameters of Degradation

To further investigate the effect of the 11S regulator on the proteasomal cleavage of substrates with oligoglutamine fragments, we determined the kinetic parameters of their degradation by proteasome complexes of different compositions. The obtained apparent *K*_m_ and *V*_max_ values are summarized in Table 2.

The highest efficiency of hydrolysis was obtained for Dabcyl-KQ10PPD-EDANS by the 26S proteasome in the presence of the 11S regulator. The addition of 11S increased the hydrolysis rate by more than two-fold; in contrast, the cleavage efficiency of Dabcyl-KQ5GD-EDANS by 26S did not change in the presence of the 11S regulator. In comparison to the two other peptides, the hydrolysis of Dabcyl-KQ10GD-EDANS proceeds at a slower rate, especially in the case of the 26S proteasome. This fact, together with the low solubility of this peptide, makes it the worst substrate in terms of cleavage efficiency.

### 2.4. Determination of Proteolysis Sites

Next, we determined proteolysis sites for our oligoglutamine-containing peptides using liquid chromatography-mass spectrometry (LC-MS). The chromatogram of the Dabcyl-KQ5GD-EDANS solution in DMSO (Figure 3a) has a peak at the retention time of 1.69 min. According to the mass spectrum (Figure 3b), this peak corresponds to the Dabcyl-KQ5GD-EDANS (M = 1458.69 g/mole; z = 2; m/z = 729) molecule. Another fragment with m/z = 252 corresponds to the Dabcyl- (C15H14N3O), and the one with m/z = 380 corresponds to the DabcylK- (C21H26N5O2) fragment. These peaks are likely to arise due to fragmentation in a mass spectrometer since, upon hydrolysis, the masses of these fragments would be 269 and 397, respectively. After the incubation with proteasome for 48 h, the peak at 1.69 min disappeared (Figure 3c), and two other peaks appeared with retention times equal to 1.45 and 1.72 min. These peaks were assigned to the N- and C-ends of the peptide on the basis of the difference in the absorption spectra of the Dabcyl and EDANS moieties. The first peak with m/z = 284.89 (z = 2) had an absorption maximum at 320 nm, corresponding to the protonated QGD-EDANS moiety (Figure 3e), and the second peak with m/z = 654 (z = 1) had an absorption maximum at 480 nm, corresponding to the Dabcyl-KQ2 fragment (Figure 3d). Other products were not detected either at short (15 min) or longer hydrolysis times (72 h). Hence, we showed that the proteasome is able to hydrolyze a peptide bond within a short oligoglutamine fragment.

Next, we identified cleavage sites for the substrate with a longer oligoglutamine fragment (Figure 4). The peak with a retention time of 1.06 min (Figure 4a) that corresponds to the Dabcyl-KQ10PPD-EDANS (M = 2236.38 g/mole; z = 2; m/z = 1118) molecule (Figure 4b) disappeared after the substrate was incubated with the 20S proteasome for 48 h. Instead, two other peaks appeared, at 1.14 and 1.37 min (Figure 4c). According to UV spectra for both peaks, the fragment at 1.14 min had Dabcyl chromophore, and the second fragment carried EDANS fluorophore. Finally, according to mass calculation, the first product was attributed to the Dabcyl-KQ2 fragment (Figure 4d) and the second one to the Q7PPD-EDANS moiety (Figure 4e).

After 48 h of the reaction, the starting substrate was almost fully degraded. To make sure that no other reaction products were present at the earlier stages of the process, we analyzed the same reaction mixture after 2 h of the reaction. The same peaks were observed on the chromatogram, but the product peaks were of smaller height, and the peak of the starting substrate was also detected. No other products were present.

Next, we analyzed the time course of Dabcyl-KQ10PPD-EDANS hydrolysis by different proteasome complexes using aliquot characterization. The degradation by both 20S and 26S went faster in the presence of the 11S regulator at the initial stage of the reaction during the first hour of hydrolysis (Figure 5). These results coincide with our kinetic findings that 11S accelerates hydrolysis of Dabcyl-KQ10PPD-EDANS not only by the 20S proteasome but also by the 26S proteasome.

To propose a mechanism for the hydrolysis of the Gln-Gln peptide bond by the proteasome, we docked Dabcyl-KQ5GD-EDANS into catalytic constitutive β1, β2, and β5 as well as immune β1i, β2i, and β5i subunits using the structure of the mouse proteasome (PDB: 3UNE for constitutive and 3UNH for immunoproteasome [32]) and AutoDock Vina™ (Figure 6). The Thr1 residue was chosen as the center of a 30 × 30 × 30 Å cube, which is the double size of the substrate’s length. 90 rounds of docking were carried out for each subunit, and the most frequent productive binding of the substrate (an arrangement of the substrate in which any of the peptide bonds is located near the catalytic residue Thr1 in the correct direction so that its hydrolysis is possible) was observed for subunits beta1, beta2, and beta2i. The peptide backbone of the substrate is fixed by multiple hydrogen bonds in almost all subunits, but the side chain entering the S1 binding pocket can form multipoint binding only in these three catalytic subunits. The substrate arrangement inside the proteolytic chamber leads to the hydrolysis of the peptide bond between Gln2 and Gln3 only in beta-1, beta-2, and beta-2i catalytic subunits. There are several hydrogen bonds that drive substrate binding in the active site, including the bond between the side chains of Gln2 (Gln3) and Arg45 (Gly45) at the S1 pocket and the bond between the carbonyl oxygen of Gln2 (Gln3) and the hydroxyl of Thr21 (Ser21). Based on this docking data, we propose that proteasomal hydrolysis between two Gln residues is possible but preferably takes place at the beginning of the oligoglutamine sequence (mostly after the second or third Gln residue). This result coincides with our experimental findings.

### 2.5. The Identification of Proteasome Catalytic Subunit That Digests Oligoglutamine Substrates

To investigate which proteasomal subunit can hydrolyze the Gln-Gln bond, we examined the effect of various proteasome inhibitors on the rate of hydrolysis of Dabcyl-KQ5PPD-EDANS and Dabcyl-KQ10PPD-EDANS, as well as other fluorogenic substrates that are widely used in the measurements of different types of proteasomal activity (Table 3, Figure 7).

It was found that the selective inhibitor of caspase-like activity, Z-P-nLeu-D-CHO, does not affect the rate of hydrolysis of oligoglutamine substrates, while the inhibitors of chymotrypsin-like activity reduce the rate of hydrolysis of such substrates only at high concentrations. Marizomib [33,34], an inhibitor with β-lactone-γ-lactam active warhead, had different IC50 values for different types of proteasome specificity and, therefore, was chosen to identify proteasome catalytic subunits that are responsible for the hydrolysis of glutamine-glutamine bonds. Since IC50 values for oligoglutamine substrates were between IC50 values for trypsin-like and caspase-like types of specificity, several types of catalytic subunits might be involved in hydrolysis. Also noteworthy is the wide range of concentrations at which the inhibition of oligoglutamine substrates was observed (Figure 7), in contrast to standard substrates. Thus, Hill coefficients differed between two groups of substrates, with the inhibition of the hydrolysis of oligoglutamine substrates by marizomib showing a negative cooperative effect.

## 3. Discussion

In this study, we demonstrated that proteasomal degradation of oligoglutamine peptides of different lengths results in short fragments cleaved from the N-end of the initial peptide. We showed that the addition of regulatory subunits 11S and 19S accelerates the rate of hydrolysis but fails to change the site of proteolysis (both by the 26S proteasome and by the 19S-20S-11S complex). Both 20S and 26S proteasomes and 11S regulator were isolated from the murine brain, which is the most relevant tissue since mutant polyglutamine proteins accumulate in patient brains. The absence of other proteases in isolated 20S and 26S proteasomes was confirmed by the inhibitory analysis with bortezomib and epoxomicin; both inhibitors showed IC50 values similar to previously published data for proteasomes from mouse tissues [35]. To distinguish between the 20S and 26S proteasome complexes, we used the SDS test: in the presence of 0.02% SDS, the 20S proteasome was activated and the 26S proteasome was inhibited (Appendix A). The presence of different catalytic subunits was verified by Western blot (Appendix A). The presence of the 26S proteasome was also confirmed by native electrophoresis (Appendix A). The absence of a fluorescent band at the level of the 20S proteasome in the 26S sample indicates that it only contains the 26S complex, and the diffused band is likely to reflect the presence of both single- and double-capped 26S particles (19S-20S and 19S-20S-19S).

First, we designed new internally quenched peptide substrates containing an oligoglutamine fragment and a FRET pair. The addition of Pro residues reduced the aggregation of the oligoglutamine peptide, which is consistent with another study showing that the sequence of at least six prolines (P6) is required to prevent the aggregation of Q40 [36]. In our work, we used the Q10 sequence with two prolines. The distances between EDANS fluorophore and Dabcyl quencher were calculated using the Scratch Protein Predictor structure prediction tool and proved to be sufficient to provide energy transfer. The amplification of the fluorescence signal after complete substrate hydrolysis supports our theoretical calculations. Table 1 and Appendix A show the characteristics of the designed substrates. It should be noted that the substrate with five glutamines is degraded with the highest efficiency due to the largest increase in fluorescence during hydrolysis. Although the Dabcyl-KQ10PPD-EDANS substrate is longer than Dabcyl-KQ10GD-EDANS and the increase in fluorescence signal was slightly lower (although sufficient), this peptide is more soluble, and its sequence better represents the amino acid sequence of huntingtin (which has a polyproline tract after the polyglutamine region). Therefore, we chose it as a model substrate for studying the hydrolysis of oligoglutamine-containing peptides by the proteasome. A long oligoglutamine peptide with Q40 or more glutamine residues will be very harshly soluble and also have a too long distance between fluorophore donor and acceptor, and this will be the limiting factor for FRET energy transfer, making such a peptide impossible to use.

The specific activity of the 26S proteasome with both oligoglutamine-containing substrates was much higher than that of the 20S proteasome, reflecting the ability of the 19S regulator to open the entrance to the proteolytic chamber of the 20S core. The shorter oligoglutamine-containing substrate Dabcyl-KQ5GD-EDANS showed an increase in hydrolysis rate in response to a growing concentration of 20S+11S complexes, but this was not observed for hybrid 26S+11S complexes. This result is consistent with our previous data obtained on short-model substrates [29]. Conversely, the rate of degradation of the longer oligoglutamine-containing substrate Dabcyl-KQ10PPD-EDANS, consisting of 10 glutamine residues (a total of 14 amino acid residues), by both 20S and 26S proteasomes markedly increased upon the addition of the 11S regulator in a concentration-dependent manner (Figure 2a,b). Proteasomes are dynamic structures, and the equilibrium between free 20S particles and 20S complexes with 19S or 11S regulators is reversible [3,37]. The increase in 26S proteasome activity in the presence of the 11S regulator most likely results from the formation of hybrid 26S+11S complexes. The 26S proteasome is a complex of the 19S regulator protein and the 20S proteasome, and the 11S protein can be associated with the “single-capped” 26S [30], resulting in a hybrid 19S-20S-11S structure. In the case of shorter substrates, the products of degradation are already small and can easily exit the internal catalytic space of a proteasome. On the other hand, the degradation of longer substrates can lead to longer products, and we hypothesize that the 11S regulator might assist in their departure from the proteasome. Thus, ubiquitinated Htt can be recognized by the 19S regulator attached to one end of the 20S cylinder, while the 11S regulator, associated with another end of the 20S cylinder, can promote a faster release of degradation products from the proteolytic chamber. A key question is which feature contributes to the increased 26S activity towards longer substrates: the length of the oligoglutamine fragment or the length of the whole peptide. To address it, we examined the changes in the rate of degradation of two peptide substrates, the standard proteasome substrate Suc-LLVY-AMC and the fortuitous substrate RE(EDANS)SQNYPIVQK(Dabcyl)R, by 20S and 26S proteasomes after adding the 11S regulator. The “short peptide” was the common fluorescent substrate for proteasome assays, and the HIV-protease substrate was the longer substrate (Figure 2c,d). These two substrates were hydrolyzed more efficiently by the 20S+11S complex than by the 20S proteasome alone; the reaction rate increased in a concentration-dependent manner. Moreover, there was a remarkable difference in the effect of the 11S regulator on 26S proteasome activity: the regulator enhanced degradation of the long HIV-protease substrate RE(EDANS)SQNYPIVQK(Dabcyl)R, but not in the case of the short model tetrapeptide substrate Suc-LLVY-AMC. Summing up all the data, one may conclude that the increased degradation of longer peptides by the 26S+11S hybrid complexes, compared to the 26S, is mainly associated with peptide length. Meanwhile, the addition of the 11S regulator to the 26S proteasome has no effect on the degradation of shorter peptides. 

To thoroughly investigate the impact of the 11S regulator on the proteasomal degradation of substrates carrying oligoglutamine fragments, we determined the kinetic constants of hydrolysis of these modified peptides by the 20S proteasome as well as different proteasome complexes. To ensure that the conditions of each experiment were the same, we used a constant 1:4 molar ratio between the proteasome and 11S regulator protein. The Michaelis–Menten equation was used to describe kinetic experiments with enzymes. Although it does not reflect the actual enzyme kinetic mechanism in our reactions, the Michaelis–Menten equation can still be used to calculate kinetic parameters for further comparison. The obtained values for apparent Km and Vmax (Table 2) agreed with our previous findings that the 11S regulator enhances 20S proteasome activity for both short and longer substrates and increases 26S proteasome activity only for longer substrates. The defined apparent Michaelis constants were in the typical range (5–65 µM) for proteasomal peptide substrates. The presence of prolines in Dabcyl-KQ10PPD-EDANS greatly contributed to peptide solubility and made it a better substrate for the proteasome than Dabcyl-KQ10GD-EDANS, since the latter demonstrated the lowest apparent Km values among all three peptides. Our data analysis presented in Table 2 shows that for all three substrates, the apparent Michaelis constants did not change substantially after the addition of the 11S regulator, thus indicating that 11S had no effect on the substrate affinity for the 20S or 26S proteasome but increased the peptidase activity of 20S for all substrates and of 26S for longer substrates.

A distinct feature of FRET-containing substrates is that the fluorescence signal increases during the hydrolysis of any peptide (amide) bond within the substrate. Therefore, to determine the exact proteolysis site, one needs to separate the resulting fragments and determine their composition. Using the LC-MS technique, we identified that the Dabcyl-KQ5GD-EDANS substrate was predominantly cleaved at the Q2-Q3 and Q4-Q5 bonds by all the proteasome complexes studied in our experiments (20S, 20S+11S, 26S, and 26S+11S), yielding Dabcyl-KQ2 and QGD-EDANS fragments that were identified in reaction mixtures (Figure 3 and Appendix A). Both single Gln and Gln-Gln dipeptides cannot be detected using LC-MS due to their high hydrophilicity, absence of optical absorption in the 250–900 nm range, and low ionizability [19]. A longer substrate, Dabcyl-KQ10PPD-EDANS, also demonstrated the digestion pattern indicative of Gln-Gln endopeptidase activity, with Dabcyl-KQ2 and Q7PPD-EDANS being the main products of its digest (Figure 4). These products correspond to the cleavage of Q2-Q3 and Q3-Q4 bonds in the substrate. This experiment was performed several times, but there were no products of digestion in the middle or near the C-end of the oligoglutamine fragment. These results show that the hydrolysis takes place at two sites near the N-end of the oligoglutamine fragment and indicate that the proteasome cannot hydrolyze peptide bonds in the middle of the oligoglutamine fragment, even in the presence of regulator proteins. However, it remains possible that hydrolysis continues and other products can appear after 72 h of reaction or later. It is generally considered that there are three amino acid residues in the bonding zone in front of the hydrolyzable bond [38]. Thus, if we saw the hydrolysis after KQ3, then all three positions would be occupied by glutamine residues, and if nothing interfered with hydrolysis anywhere, it would occur. However, we did not notice hydrolysis near the C-end of the Q10 peptide. Another interesting observation is that the composition of the proteasome complex had no effect on the cleavage site since all identified products of hydrolysis were identical for all proteasome complexes (20S, 20S+11S, 26S, 26S+11S) (Appendix A). Time-course analysis of Dabcyl-KQ10PPD-EDANS hydrolysis by different proteasomes (Figure 5) showed that the addition of 11S significantly accelerated the reaction during its first hour, an observation that supports our kinetics data. Our results confirm and extend previous observations [11] about the ability of proteasomes to cleave only within the initial few glutamines of the polyQ fragment. 

Additionally, the short Dabcyl-KQ5GD-EDANS substrate was analyzed by docking studies in the mouse proteasome (Figure 6). β1, β2, and β2i subunits provided the most favorable binding modes between the oligoglutamine fragment and uncharged hydrophilic side chains in the hydrophilic substrate binding pockets of these three catalytic subunits. The peptide chain is well incorporated into the substrate-binding channel, positioning the Gln2 or Gln3 residue of the substrate in the S1 pocket of the active site since its amide group favorably interacts with Arg45. We assume that this dominant docking position can be attributed to the formation of multiple hydrogen bonds between the substrate and the enzyme. However, it is still possible that the fluorophore and quencher groups can additionally fix this docked pose via hydrophobic or stacking interactions. Docking is highly consistent with biochemical data, thus providing additional confirmation to the results obtained by the LC-MS technique.

To examine the role of the proteasome in polyQ degradation in more detail, an inhibition study was carried out. The two most commonly used inhibitors, Z-LLL-CHO (MG132) and Bortezomib, inhibited proteasomal hydrolysis of oligoglutamine substrates only at fairly high concentrations with IC50s in the micromolar range (Table 3), and the specific inhibitor of caspase activity did not affect the rate of hydrolysis. According to half-inhibition concentrations (IC50) obtained for marizomib, a mixed type of hydrolysis takes place for both oligoglutamine substrates (participation of active centers with caspase-like and trypsin-like specificity at the same time). It is interesting to note the change in Hill coefficients between two groups of substrates, oligoglutamine substrates and standard peptides (Figure 7). For standard substrates, this value is 1.1–1.5, which indicates a positive cooperative binding effect. Since the proteasome has two active sites for each type of substrate specificity, and model substrates strictly correspond to only one type, complete blocking of the hydrolysis of such substrates requires the binding of two inhibitor molecules, one in each active site. In the case of oligoglutamine substrates, the Hill coefficient was 0.4–0.7. A negative cooperative effect is observed, with an apparent decrease in the affinity for the inhibitor when the active centers are saturated. It is known that negative cooperativity can result from the heterogeneity of binding sites, i.e., the hydrolysis of oligoglutamine substrates is simultaneously carried out by different types of proteasome active centers. The trypsin and caspase catalytic centers are adjacent and located nearby in the proteolytic chamber, which can explain the mixed type of specificity observed in our experiments. It has been shown that in the nervous system, 11Sαβ is upregulated upon stress, suggesting a protective role of 11Sαβ against proteostasis dysfunction [39], but unlike the 11Sγ regulator, which activates trypsin activity to the greatest extent [40], 11Sαβ non-selectively stimulates total peptidase activities of the 20S proteasome [41].

## 4. Materials and Methods

### 4.1. Peptide Substrates and Inhibitors

Peptide FRET substrates with ten glutamine residues were procured from GenScript (Piscataway, NJ, USA); the substrate with five glutamine residues was purchased from Peptide Protein Research Ltd. (Fareham, UK). The standard fluorescent substrates for proteasome assays (N-succinyl-Leu-Leu-Val-Tyr-7-amino-4-methylcoumarin, Suc-LLVY-AMC) and HIV protease substrate 1 (Arg-Glu(EDANS)-Ser-Gln-Asn-Tyr-Pro-Ile-Val-Gln-Lys(Dabcyl)-Arg) were from Sigma-Aldrich (St. Louis, MO, USA) and Invitrogen (Carlsbad, CA, USA), respectively. Ac-RLR-AMC and Z-LLE-AMC were from UBPBio (Dallas, TX, USA). Bortezomib was from LC Laboratories (Woburn, MA, USA), MG-132 (Z-LLL-CHO), leupeptin (as hemisulfate salt), and marizomib was from Sigma-Aldrich, and Z-P-nLeu-D-CHO was from Enzo Life Science (Farmingdale, NY, USA). All other reagents were of the highest quality available or for molecular biology grades and were used without additional purification.

### 4.2. Isolation of Proteasome Particles

The 20S and 26S proteasomes were isolated using the procedure described in [42,43,44]. All the chromatographic stages were performed using an HPLC system (AKTA Purifier, GE Healthcare, Uppsala, Sweden). Chromatographic profiles are shown in Appendix A. A protein kit for size-exclusion chromatography with a mass range of 65 to 669 kDa (Sigma-Aldrich) was used to calibrate the Superdex 200 column, and blue dextran was used to determine the dead volume of the column. Isolation of the 11S regulatory protein was carried out identically to the procedure used for the 20S proteasome, with the following modifications [29]: leupeptin was added to the buffer for the homogenate (50 mM Tris-HCl, 100 mM NaCl, 1 mM EDTA, 1 mM DTT, 10% glycerol, pH = 7.5) to provide carboxypeptidase inhibition; salting out was conducted using (NH_4_)_2_SO_4_ under 85% saturation. Ion-exchange chromatography was performed using a DEAE Toyopearl column (Tosoh bioscience, San Francisco, CA, USA) equilibrated with TSD buffer (20 mM Tris-HCl, 1 mM EDTA, 1 mM DTT, 10% glycerol, pH 7.5) and eluted with a NaCl concentration gradient from 0 to 0.3 M; size-exclusion chromatography was performed on the Superdex 200 column equilibrated with TSD buffer. Fractions in which proteins approximately corresponded to MW 200 kDa (11S is a heptamer consisting of 28 kDa monomers) were collected and analyzed. After each chromatography run, the collected fractions were analyzed for the presence of the 11S protein using the fluorometric method by increasing the activity of the 20S proteasome towards a specific fluorescent substrate (Suc-LLVY-AMC). Chromatographic profiles are shown in Appendix A. The fractions containing the 11S protein were transferred into TSD buffer by dialysis (overnight at +4 °C). Dialyzed 11S protein was concentrated using Amicon Ultra 10K (Millipore, Cork, Ireland), and then protein concentration and 20S proteasome activation ability were measured. The 11S regulator itself had no peptidase activity. 

The fact that protease activity of the purified product is provided only by 20S and 26S proteasomes was confirmed by inhibitory analysis using at least three concentrations of bortezomib (20 and 500 nM) and MG-132 (100 and 1000 nM) proves that the 20S proteasome was in closed gated form and to distinguish between 20S and 26S proteasome complexes. In the presence of 0.02% SDS, the activity of the 20S proteasome increased by 13–15 times, and the 26S proteasome was completely inhibited. Dot blotting of the fractions containing the 20S or 26S proteasome and the 11S particle was performed according to the standard procedures using the appropriate primary antibodies (all primary antibodies were from Enzo Life Sciences, Inc., Farmingdale, NY, USA) and secondary anti-species antibodies conjugated to horseradish peroxidase (secondary anti-species antibodies were from Sigma-Aldrich), in accordance with the manufacturer’s instructions. Imaging was carried out using a ChemiDoc scanner (BioRad, Hercules, CA, USA) with 15–30 min of exposure. Protein concentration was determined by absorbance at 280 nm using a NanoDrop-2000 spectrophotometer (Thermo Scientific, Waltham, MA, USA), according to the Bradford technique using apoferritin for plotting a calibration graph and also with the Lowry technique using BSA for plotting a calibration graph (BCA assay, Thermo Scientific, Rockford, IL, USA). The final concentrations of concentrated solutions were found to be approximately equal to 0.5 µM for the 20S proteasome, 0.2 µM for the 26S proteasome, and 2 µM for the 11S regulator protein.

### 4.3. Kinetic Measurements

To determine the dependence between the degradation rates and the 11S protein-to-proteasome molar ratio, substrate solution in DMSO was mixed with buffer D with pH = 7.5 (20 mM Tris-HCl, 1 mM EDTA, 1 mM DTT; also 5 mM MgCl_2_, 1 mM ATP for measurements with the 26S proteasome) to a final concentration of 50 µM. The FRET substrate concentration in DMSO was determined before each experiment by measurement of the optical density of centrifuged substrate solution at 473 nm using the molar absorption coefficient of 32,000 [L·mol^–1^·cm^–1^]. To achieve better substrate dissolution, the percentage of DMSO in the final solution was 2%. To ensure that the substrate was completely dissolved, the substrate solution in Tris-HCl buffer with 2% DMSO was centrifuged for 5 min at 13,400 rpm (MiniSpin, Eppendorf, Hamburg, Germany) before each assay. Substrate solution portions (90 µL) were placed in a black opaque 96-well plate (Greiner Bio-One GmbH, Frickenhausen, Germany), and 5 µL of 20S proteasome (or 10 µL of 26S proteasome) was then added. A different volume of 11S protein was added to each well in order to obtain the following molar ratios between 20S or 26S proteasome and 11S protein: 1:0, 1:1, 1:2, 1:4, 1:5, 1:8, and 1:10. The total volume per well was adjusted to 110 μL using buffer D. The fluorescence intensity was measured on a Multilabel Reader VictorX5 microplate fluorimeter (PerkinElmer, Waltham, MA, USA) for 1 h at measurement intervals of 30 s and at a temperature of 37 °C. Excitation and fluorescence wavelengths were 340 nm and 490 nm, respectively, for FRET substrates and 355 nm and 460 nm for the AMC substrate. The measurement results were processed in MS Excel. The hydrolysis rate was calculated using the formula (Equation (2)):(2)$$\frac{\frac{dI}{dt}\times [S]}{I^{100}-I^{0}}$$
where *dI*/*dt* is the change in fluorescence intensity over the appropriate time; [*S*] is the substrate concentration; *I*^100^ is the fluorescence intensity of EDANS (or AMC) at an appropriate concentration; and *I*^0^ is the initial fluorescence intensity of the substrate without proteasome addition.

### 4.4. Determination of Apparent Kinetic Parameters

Enzyme reactions were initiated by adding 5 µL of 20S proteasome, 10 µL of 26S proteasome, or 5 µL of 20S proteasome +5 µL of 11S regulator, or 10 µL 26S proteasome +4 µL of 11S regulator to 90 µL of each substrate, with their concentrations ranging from 5 to 100 µM. The fluorescence intensity was measured for 1 h at 30-s intervals at an excitation wavelength of 340 nm, a fluorescence wavelength of 490 nm, and a temperature of 37 °C. To additionally check that the FRET-substrate does not precipitate, the absorption spectra (250–550 nm) of the substrate solution were recorded after 1 h, 4 h, and 24 h of incubation at 37 °C, and no changes were observed in the absorption spectrum. The apparent kinetic constants were calculated through the Michaelis–Menten method using GraphPad Prism 7 software.

### 4.5. Determination of Cleavage Sites by LC/Mass Spectrometry

Peptide samples were analyzed on an Acquity UPLC system (Waters, Milford, MA, USA) connected to a PDA detector and TQD mass spectrometer (Waters). The column was 2.1 × 50 mm Acquity BEH C18, 1.7 µm (Waters). The gradient program was as follows: from 5%B to 50%B per 3 min, from 50%B to 5%B per 0.1 min, at a flow rate of 0.5 mL/min and a temperature of 35 °C. Solvent A: 20 mM formic acid in acetonitrile/water (1:20)/Solvent B: 20 mM formic acid in acetonitrile. Instrument settings and measuring parameters TQD were as follows: capillary voltage 3.5 kV; cone voltage 25 V; source temperature 120 °C; desolvation temperature 450 °C; cone gas flow 50 L/h; desolvation gas flow 800 L/h; MS1 scanning mass from 50 to 1500 Da; positive polarity. The PDA detector range was 220–500 nm. The resulting raw data were analyzed using the MassLynx software (version 4.1 SCN919).

### 4.6. Time-Course of Dabcyl-KQ10PPD-EDANS Cleavage by Different Proteasome Complexes

In a series of reactions, 80 μL of 20S or 26S proteasomes (0.1 and 0.24 mg/mL, respectively) were mixed with 1.8 μL of a substrate solution in DMSO (5 mM) and 40 μL of an 11S regulator solution with a protein concentration of 0.12 mg/mL, and the volume was made up to 180 μL with freshly prepared buffer (50 mM Tris-HCl, 1 mM EDTA, 1 mM DTT, 10% glycerol, +5 mM ATP-MgCl2 in the case of 26S, pH = 7.5). In the control solution, 40 µL of 11S solution, 1.8 µL of substrate solution, and 138.2 µL of buffer were mixed. The mixtures were thermostated at 37 °C, and samples were taken in 20 µL at 0; 15; 30; 60 min; 2.5 h; 24 h; 48 h; and 72 h. 1 μL of bortezomib (1mM in DMSO) was added and frozen. Samples were analyzed by HPLC using the column 2.1 × 50 mm Acquity BEH C18, 1.7 µm, and Acquity UPLC systems (Waters). The area of the peaks was calculated using supplier MassLynx software (version 4.1 SCN919). To additionally check that the FRET-substrate does not precipitate, the absorption spectra (250–550 nm) of the substrate solution were recorded after 1 h, 24 h, and 48 h of incubation at 37 °C, and no changes were observed in the absorption spectrum.

### 4.7. Docking Study

Docking was carried out using the AutoDock Vina 1.1.2 program. The ligand structure was drawn using the MarvinSketch v16.12.19 program. Docking was carried out at the catalytic sites of all six catalytic subunits (β1c, β2c, β5c, β1i, β2i, β5i) using a full proteasome structure; a 30 × 30 × 30 Å cube with a center located midway between oxygen and carbon atoms of the side chain Thr1 was chosen as the docking cell. The structures were taken from the PDB database (PDB ID 3UNE for constitutive and 3UNH for immunoproteasome), and 90 dockings were made. The docking results were analyzed using the PyMOL^TM^ V2.4.1 program.

## 5. Conclusions

The hydrolysis of three peptide substrates with five or ten glutamine residues by proteasome complexes of various compositions has been examined. The LC/mass spectrometry and docking methods were used to demonstrate that the proteasome can only cleave small fragments, two to three amino acid residues in length, from the N-end of the oligoglutamine sequence. The addition of the 11S regulator accelerates this process but does not change the hydrolysis site; consequently, the length of the cleaved fragment remains unchanged. Thus, it is possible that long undigested polyglutamine fragments leave the proteasome proteolytic chamber with further accumulation in the cell. By comparing the rates of hydrolysis after the addition of 11S protein and the kinetic parameters of proteasomal degradation for each substrate, we showed that the 11S regulator enhances 20S proteasome activity for all three substrates, regardless of their length. 11S also accelerates 26S proteasome hydrolysis of substrates carrying ten glutamine residues. We propose that the 11S regulator can stimulate the removal of larger products of hydrolysis resulting from the longer substrate. Thus, modulating 26S proteasome activity in the context of proteinopathic diseases, either by phosphorylation [45,46] or in an 11S-dependent manner [47], represents a promising area for future research.

## Figures and Tables

**Figure 1 ijms-24-13275-f001:**
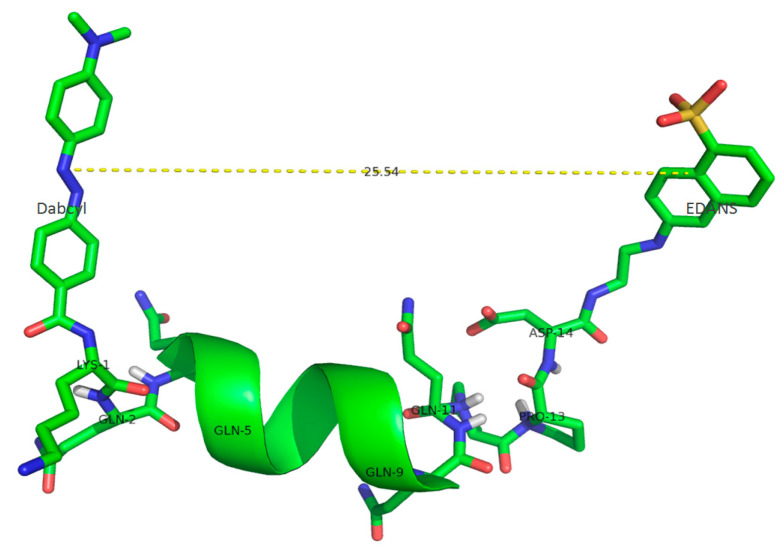
The predicted and visualized structure of a Dabcyl-KQ10PPD-EDANS molecule.

**Figure 2 ijms-24-13275-f002:**
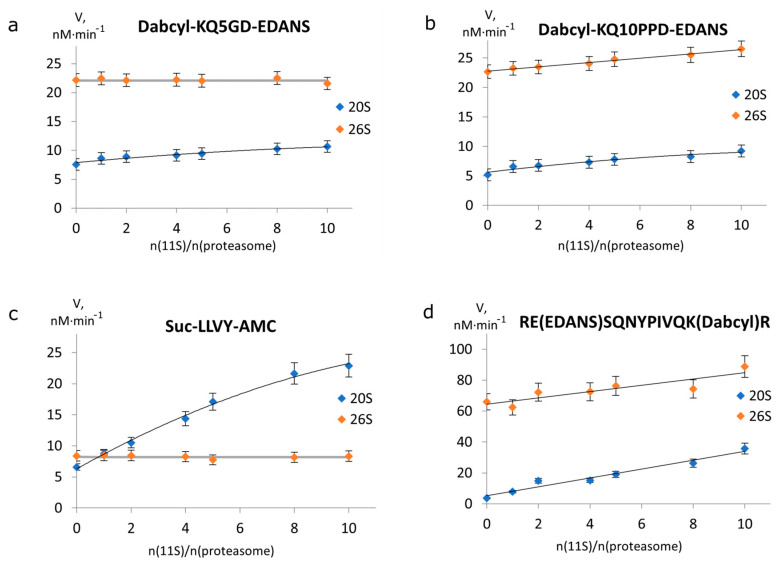
Degradation rates of various peptide substrates at different molar ratios between the 11S regulator protein and the proteasome. Kinetic experiments were performed with peptide substrates containing (**a**) oligoglutamine fragments of five residues; (**b**) oligoglutamine fragments of ten residues; (**c**) standard short substrate; and (**d**) long HIV-protease substrate [S]. [20S] = 14.2 nM, [26S] = 13.8 nM, for all substrates [S] = 50 µM, in 50 mM Tris-HCl buffer pH = 7.5, containing 1 mM EDTA, 1 mM DTT, and 1% DMSO (with addition of 5 mM MgCl2, 1 mM ATP for 26S proteasome).

**Figure 3 ijms-24-13275-f003:**
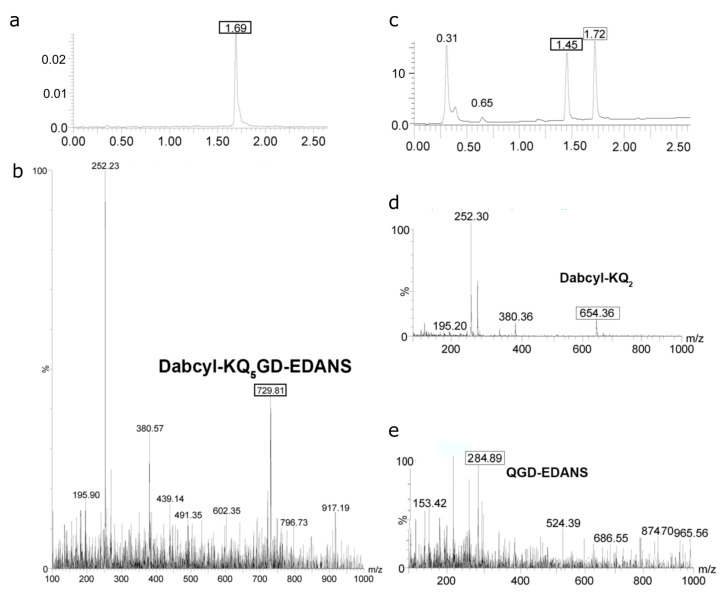
Cleavage sites of Dabcyl-KQ5GD-EDANS by the 20S proteasome. (**a**) Chromatogram of substrate solution in DMSO, detection by UV-VIS. (**b**) Mass spectrum of the peak at 1.69 min. (**c**) Chromatogram of substrate mixture with 20S proteasome after incubation for 48 h, UV-VIS detection. (**d**) Mass spectrum of the peak at 1.72 min. (**e**) Mass spectrum of the peak at 1.45 min.

**Figure 4 ijms-24-13275-f004:**
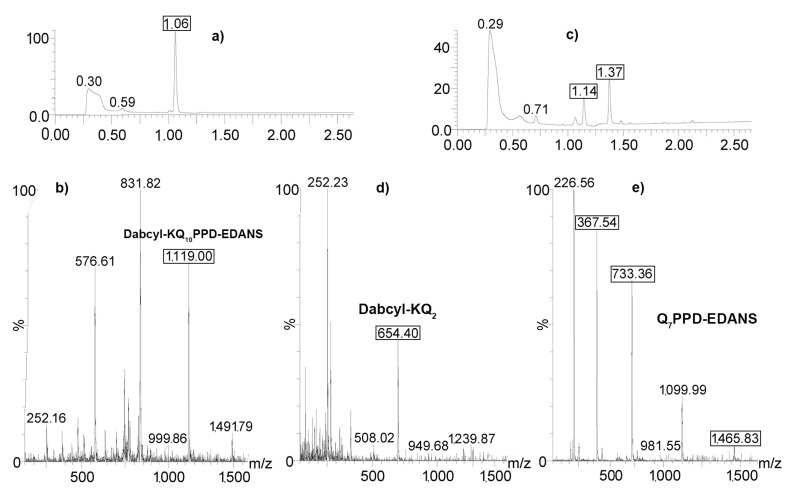
Cleavage sites of Dabcyl-KQ10PPD-EDANS by the 20S proteasome. (**a**) Chromatogram of substrate solution in DMSO. (**b**) Mass spectrum of the peak at 1.06 min. (**c**) Chromatogram of substrate mixture with 20S proteasome after incubation for 48 h. (**d**) Mass spectrum of the peak at 1.14 min. (**e**) Mass spectrum of the peak at 1.37 min.

**Figure 5 ijms-24-13275-f005:**
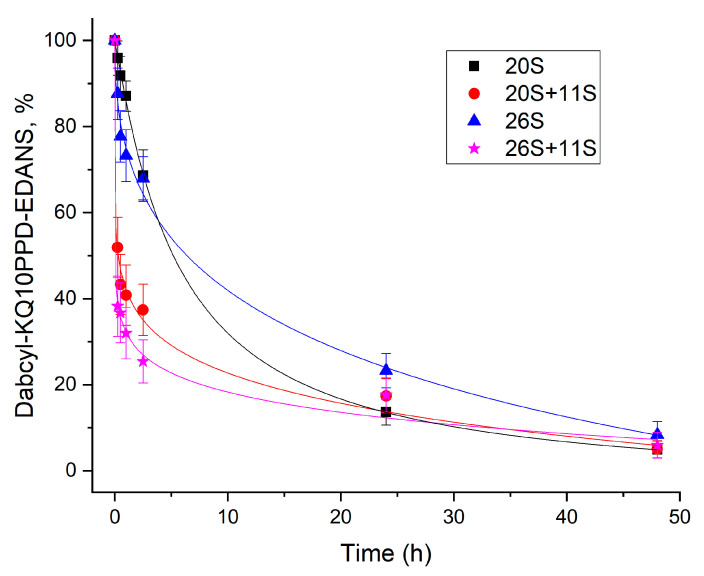
Time-course of Dabcyl-KQ10PPD-EDANS cleavage by different proteasome complexes. Aliquots were taken from the reaction mixture ([20S] = 50 nM or [26S] = 45 nM, [11S] = 200 nM, [S] = 200 µM, 37 °C) after 0 min, 15 min, 30 min, 1 h, 2.5 h, 24 h, and 48 h.

**Figure 6 ijms-24-13275-f006:**
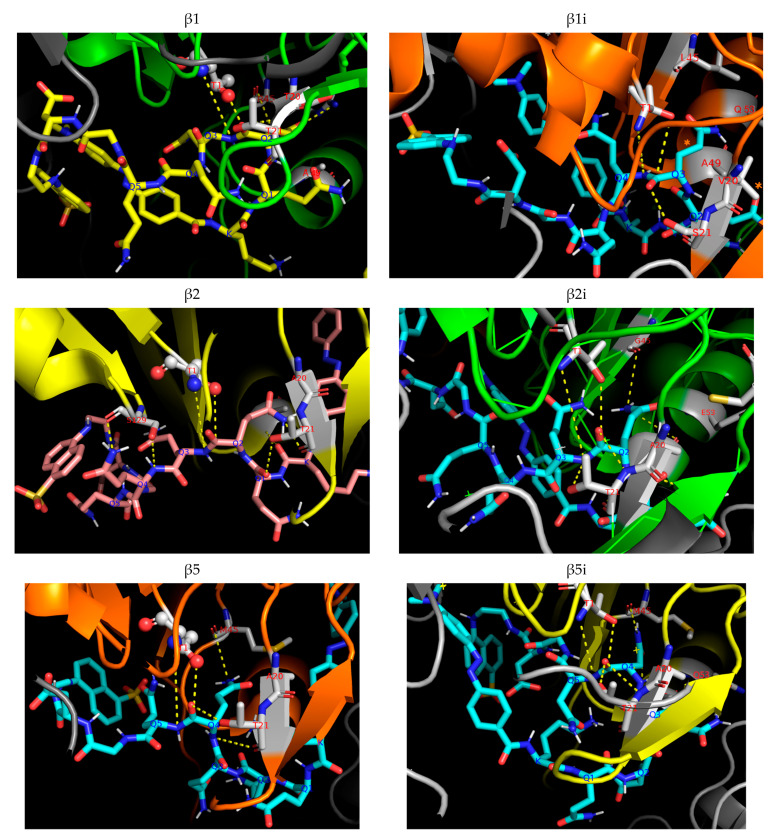
The binding mode of Dabcyl-KQ5GD-EDANS in six proteasome catalytic subunits The active site is shown as a cartoon representation; the substrate is shown using the stick representation; the substrate’s Gln residues are marked in blue at Calpha atom. Thr1 residue is shown in gray; residues of substrate binding pockets are marked with one-letter code and residue number in red. Hydrogen bonds are shown in yellow. The figure was prepared using PyMOL software V2.4.1.

**Figure 7 ijms-24-13275-f007:**
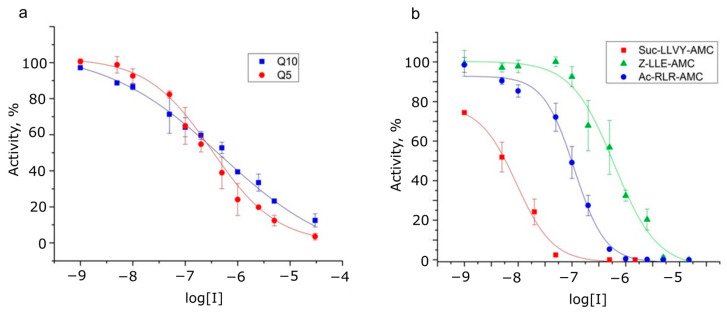
The dependence of 20S proteasome activity on marizomib concentration during hydrolysis of peptide fluorogenic substrates. (**a**) hydrolysis of model oligoglutamine substrates, Dabcyl-KQ5-PPDE-EDANS (Q5) or Dabcyl-KQ10-PPDE-EDANS (Q10). (**b**) hydrolysis of standard coumarine-modified peptide substrates. The experimental data are approximated by the function y = A1 + (A2 − A1)/(1 + 10^(LOGx0–x)p^), where A1 and A2 are the minimum and maximum values of log10(Ci), respectively, LOGx0 is the logarithm of the EC50 (the concentration giving a response half-way between the upper and lower asymptotes), and p is the Hill coefficient.

**Table 1 ijms-24-13275-t001:** Characteristics of the FRET-peptide substrates.

Peptide Sequence	Mr/Z Calculated/Detected	*r*, Å	FRET Efficiency (E), %
Dabcyl-KQ_5_GD-EDANS	729.34/729.81 (Z = +2)	20.0	95.3
Dabcyl-KQ_10_GD-EDANS	700.31/700.7 (Z = +3)	24.9	84.4
Dabcyl-KQ_10_PPD-EDANS	1118.51/1119.24 (Z = +2)	25.5	82.4

**Table 2 ijms-24-13275-t002:** Summarized the kinetic parameters of hydrolysis of substrates containing oligoglutamine fragments of various lengths by different proteasome complexes. The fourth column shows the ratio between the Vmax of degradation by the complex of the 20S (26S) proteasome with the 11S regulator and the Vmax of degradation by the 20S (26S) proteasome. Values are represented as means ± SEM of three independent experiments run in duplicate. The molar ratio between the proteasome and the 11S regulator protein was 1:4.

**Dabcyl-KQ_5_GD-EDANS**			
**Proteasome complex**	***K*_m_, μM**	***V*_max_·10^−2^, μM·min^−1^**	***V*_max_ (+11S)/*V*_max_**
20S	52 ± 8	2.3 ± 0.3	2.7
20S+11S	60 ± 8	6.1 ± 1.4	
26S	54 ± 10	2.0 ± 0.3	1.0
26S+11S	52 ± 10	2.1 ± 0.3	
**Dabcyl-KQ_10_GD-EDANS**			
**Proteasome complex**	***K*_m_, μM**	***V*_max_·10^−2^, μM·min^−1^**	***V*_max_ (+11S)/*V*_max_**
20S	40 ± 3	0.61 ± 0.02	1.5
20S+11S	64 ± 14	0.92 ± 0.15	
26S	18 ± 2	0.09 ± 0.01	1.4
26S+11S	38 ± 9	0.12 ± 0.03	
**Dabcyl-KQ_10_PPD-EDANS**			
**Proteasome complex**	***K*_m_, μM**	***V*_max_·10^−2^, μM·min^−1^**	***V*_max_ (+11S)/*V*_max_**
20S	8.2 ± 1.5	0.73 ± 0.03	2.3
20S+11S	5.3 ± 1.3	1.66 ± 0.07	
26S	24.5 ± 10	2.3 ± 0.2	2.0
26S+11S	20.0 ± 4.6	4.73 ± 0.35	

**Table 3 ijms-24-13275-t003:** Inhibition (IC50) of 20S brain proteasome hydrolysis of standard and oligoglutamine-containing substrates by different inhibitors.

Inhibitor	IC50 Values of Peptide Substrates Hydrolysis by 20S Brain Proteasome				
	Suc-LLVY-AMC (Substrate for Chymotrypsin-like Activity)	Ac-RLR-AMC (Substrate for Trypsin-like Activity)	Z-LLE-AMC (Substrate for Caspase-like Activity)	Dabcyl-KQ_5_ PPD-EDANS	Dabcyl-KQ_10_ PPD-EDANS
Bortezomib (chymotrypsin-like activity)	8.1 ± 0.8 nM	2.6 ± 0.7 µM	23 ± 3 nM	5.3 ± 2.1 µM	4.1 ± 2 µM
Z-LLL-CHO (MG132) (chymotrypsin-like activity)	12 ± 3 nM	2.8 ± 0.3 µM	155 ± 5 nM	50 ± 6 µM	72 ± 8 µM
Z-P-nLeu-D-CHO (caspase-like activity)	100 ± 10 µM	>150 µM	5.1 ± 0.3 µM	n.d.	n.d.
Marizomib (inhibits all 3 activities)	9.3 ± 1.5 nM	111.0 ± 8.0 nM	610 ± 40 nM	323 ± 15 nM	480 ± 20 nM
(Hill coefficient)	(−1.28)	(−1.51)	(−1.1)	(−0.69)	(−0.39)

20 mM Tris-HCl buffer pH = 7.5 with 1 mM DTT, 1 mM EDTA; [20S] = 50 nM, [Suc-LLVY-AMC] = 50 µM, [Ac-RLR-AMC] = 50 µM, [Z-LLE-AMC] = 50 µM. n.d.—not detected.

## Data Availability

The data presented in this study is available in the article.

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
