# Peer review of "Brain-Derived 11S Regulator (PA28αβ) Promotes Proteasomal Hydrolysis of Elongated Oligoglutamine-Containing Peptides"

_ijms, 2023, doi:10.3390/ijms241713275_

Round 1

Reviewer 1 Report

In the submitted manuscript „Brain Derived 11S Regulator (PA28αβ) Promotes Proteasomal Hydrolysis of Elongated Oligoglutamine-Containing Peptide” the authors report that 20S and 26S proteasomes are capable of degrading the N-terminal part of oligoglutamine fragments, and claim that PA28 significantly accelerates hydrolysis without changing the specificity.

The authors used peptide substrates Dabcyl-KQ5GD-EDANS, Dabcyl-KQ10GD-EDANS, and Dabcyl-KQ10PPD-EDANS in their experiments. Since these are relatively short polyglutamine repeats, what happens when longer substrates are used, which mimic mutant forms of genes, i.e., having 40 or more polyglutamine repeats?

Also, since Dabcyl-KQ10GD-EDANS peptide shows decreased solubility, does Dabcyl-KQ5PPD-EDANS differ from Dabcyl-KQ5GD-EDANS in cleavage efficiency measurements?

The initial experiments are not repeated with Dabcyl-KQ5PPD-EDANS, even though later experiments utilize Dabcyl-KQ5PPD-EDANS instead of Dabcyl-KQ5GD-EDANS.

With Dabcyl-KQ10PPD-EDANS used as a substrate, main cleavage sites were identified between the Q2 and Q3 or Q3 and Q4 in the substrate. However, in the publication J Biol Chem. 2008 May 9; 283(19): 12919–12925, the authors report “that proteasomes cleave repetitively anywhere within a stretch of ten glutamine residues.“ How do the authors explain the discrepancy?

In Table 3, I suggest to add column specifying which activity is blocked with which inhibitor, to make the following of the results easier.

I also suggest to additionally perform experiments with site-specific proteasomal inhibitors for all 3 activity sites (or combinations thereof) to further validate their hypothesis.

Minor issues.

Lane 17: the authors should unify writing of the PA28, and not use PA28alphabeta in title, followed by α/β11Sreg and 11Sreg in the abstract.

I recommend to use PA28 in title and abstract, and to explain the alternative names in the Introduction, followed by using PA28 throughout the text.

Figure 2. Labelling of x and y axes should be improved (orientation, gap, font size).

In Lanes 39-40 the authors state: “The actual substrate specificity of a proteasome is even broader, which allows it to hydrolyze virtually any protein inside the proteolytic chamber to short peptides or amino acids.”

Can proteasome actually hydrolyse proteins to individual amino acids?

Lane 76: HEK293 instead of HEK cells should be written.

Lane 106: what does it mean “viz. PA28gamma”?

In Figure 6, it is hard to read residue numbers due to size and colour.

The authors should also unify writing for oligoglutamine peptide (i.e., they use oligoglutamine, oligo-glutamine and oligoQ in the text).

Title of the Figure S4 “Native electrophoresis in PAAG of 20S and 26S proteasome with or without addition of 11S regulator.“ What does PAAG mean?

Acknowledgments still contain the generic text, please exchange or delete: “In this section, you can acknowledge any support given which is not covered by the author contribution or funding sections. This may include administrative and technical support, or donations in kind (e.g., materials used for experiments).”

In Figure S2 western blot results are shown poorly. Except for beta2 subunit, almost no other protein bands are nicely visible. These blots should be exchanged with the blots where more lysate was analysed.

Minor improvements needed, overall OK.

Author Response

The authors used peptide substrates Dabcyl-KQ5GD-EDANS, Dabcyl-KQ10GD-EDANS, and Dabcyl-KQ10PPD-EDANS in their experiments. Since these are relatively short polyglutamine repeats, what happens when longer substrates are used, which mimic mutant forms of genes, i.e., having 40 or more polyglutamine repeats?

Response: Unfortunately, this experiment cannot be performed with long FRET substrates since the long distance between the donor and the acceptor will be a limiting factor for energy transfer. In addition, such oligopeptides are likely to be insoluble or poorly soluble (and prone to aggregation) in aqueous solutions (Chen and Wetzel, Protein Sci 2001), and it seems that such long polyglutamine sequences are soluble only when they are part of the native protein. This reasoning has been added to the discussion.

Also, since Dabcyl-KQ10GD-EDANS peptide shows decreased solubility, does Dabcyl-KQ5PPD-EDANS differ from Dabcyl-KQ5GD-EDANS in cleavage efficiency measurements?

Response: According to our data, the cleavage efficiencies are very similar between 2 short substrates: 26S activity equals to 1.1 and 0.9 nmole/(min·µg) for Dabcyl-KQ5PPD-EDANS and Dabcyl-KQ5GD-EDANS, respectively.

 The initial experiments are not repeated with Dabcyl-KQ5PPD-EDANS, even though later experiments utilize Dabcyl-KQ5PPD-EDANS instead of Dabcyl-KQ5GD-EDANS.

Response: Since we didn`t observe the change in activity between short substrates, we didn`t repeat some of the initial experiments with Dabcyl-KQ5PPD-EDANS.

With Dabcyl-KQ10PPD-EDANS used as a substrate, main cleavage sites were identified between the Q2 and Q3 or Q3 and Q4 in the substrate. However, in the publication J Biol Chem. 2008 May 9; 283(19): 12919–12925, the authors report “that proteasomes cleave repetitively anywhere within a stretch of ten glutamine residues.“ How do the authors explain the discrepancy?

Response: In our work, the proteasome was isolated from mice brain, while the authors of mentioned paper isolated the proteasome from blood cells. It is possible that the different composition of proteasome in different cell types affects its ability to interact with polyglutamine peptides.  However, further study is required to confirm this hypothesis. At the same time, we cannot exclude the possibility that the hydrolysis within Q10 sequence still occurs but at a very slow rate meaning that the concentration of these products would be below the detection limit.

In Table 3, I suggest to add column specifying which activity is blocked with which inhibitor, to make the following of the results easier.

Response: Thank you for the suggestion, this information has been added to the Table.

I also suggest to additionally perform experiments with site-specific proteasomal inhibitors for all 3 activity sites (or combinations thereof) to further validate their hypothesis.

Response: Thank you for your suggestion. To validate our hypothesis, we used marizomib, a substance that inhibits all 3 types of proteasomal activity under high concentrations.

Minor issues.

Thank you for the comments. The following changes were made to the text.

Lane 17: the authors should unify writing of the PA28, and not use PA28alphabeta in title, followed by α/β11Sreg and 11Sreg in the abstract. I recommend to use PA28 in title and abstract, and to explain the alternative names in the Introduction, followed by using PA28 throughout the text.

Response: The writing of 11S regulator was unified, the changes were made to the text.

Figure 2. Labelling of x and y axes should be improved (orientation, gap, font size).

Response: The labelling of axes was edited.

In Lanes 39-40 the authors state: “The actual substrate specificity of a proteasome is even broader, which allows it to hydrolyze virtually any protein inside the proteolytic chamber to short peptides or amino acids.” Can proteasome actually hydrolyse proteins to individual amino acids?

Response: We removed the word amino acids from text (Lane 42). It is believed that dipeptides are the smallest products of proteasomal degradation (Kisselev et al. J of Biol Chem 1998,, Saric at al. J of Biol Chem 2004).

Lane 76: HEK293 instead of HEK cells should be written.

Response: The change was made in the text.

Lane 106: what does it mean “viz. PA28gamma”?

Response: The change was made in the text.

In Figure 6, it is hard to read residue numbers due to size and colour.

Response: Thank you for your suggestion, we improved this figure.

The authors should also unify writing for oligoglutamine peptide (i.e., they use oligoglutamine, oligo-glutamine and oligoQ in the text).

Response: The naming of peptides was unified (oligoglutamine) throughout the text.

Title of the Figure S4 “Native electrophoresis in PAAG of 20S and 26S proteasome with or without addition of 11S regulator.“ What does PAAG mean?

Response: We meant polyacrylamide gel electrophoresis (PAGE). The change was made in the text.

Acknowledgments still contain the generic text, please exchange or delete: “In this section, you can acknowledge any support given which is not covered by the author contribution or funding sections. This may include administrative and technical support, or donations in kind (e.g., materials used for experiments).”

Response: The change was made in the text.

In Figure S2 western blot results are shown poorly. Except for beta2 subunit, almost no other protein bands are nicely visible. These blots should be exchanged with the blots where more lysate was analysed.

Response: Thank you for your suggestion, we improved this figure.

Reviewer 2 Report

The manuscript by Kriachkov et al is focused on the cleavage by the proteasome of polyQ proteins. The authors aims to focus on 20S, 26S and 20S-PA28 from brain to determine this and come to the conclusion that PA28 accelerated cleavage and that cleavage by all complexes occurs at same location, namely between glutamine 2 and 3.

While the authors use a nice FRET based probe for read-out, the results are overall not convincing to this reviewer. My main concern is:

1-    that the purity of the input of the complexes used has not been shown. A CBB of and SDS-PAGE gel showing pure 20S, 26S, and 11S and better characterization of this material would be required.

2-    It is unclear what really happens when 11S is added to 20S or 26S, does reconstitution work? To what extent?

3-    The authors use a short peptide that presumably diffuse into open CP complexes and does not depend on ATPase activity and the poly glutamine stretches are very short compared to the physiological problematic poly glutamine stretches in Htt and likes.

4-    That both glutamine peptides are cleaved at smilar position does not convincingly show that cleave in middle of poly glutamine does not occur. There are too few controls and variations, the lysine and preferred binding of short peptide could influence and impact cleavage event that might be very different from stretch within larger poly peptide.

5-    Complexes were purified from brain and it is described as brain related activity. However, are the purified complexes different from complexes purified from other organs? What data show this is unique from brain? This description seems more suggestive to provide perceived connection to neuronal disease than a have shown relevance for this highly in vitro study with artificial substrates.

Based on these concern I do not find that the conclusions and interpretation is supported by the data.

The message and intention of authors come across, however substantial editing would be desired to eliminate awkward English and remove unnecessary lengthy and indirect phrasing.

Author Response

While the authors use a nice FRET based probe for read-out, the results are overall not convincing to this reviewer. My main concern is:

  • that the purity of the input of the complexes used has not been shown. A CBB of and SDS-PAGE gel showing pure 20S, 26S, and 11S and better characterization of this material would be required.

Response: The purity of isolated proteasomes, i.e. the absence of other proteases, was confirmed by inhibitory analysis and by SDS addition test, data available in Table S1. Additionally, we checked that 11S regulator itself had no peptidase activity.

  • It is unclear what really happens when 11S is added to 20S or 26S, does reconstitution work? To what extent?

Response: The proteasome is a dynamic intracellular complex and can be reformed in response to various signals  (Fabre et al. Mol Syst Biol 2015). In our case, we observed the formation of 20S-11S complex when 11S was added to 20S proteasome (Figure S4 with native electrophoresis), however, it is difficult to assess to what extent the complex is forming, since the conditions in native electrophoresis and in vitro during the reaction cannot be considered identical. In the case of 26S proteasome, the formation of a hybrid complex is confirmed by indirect data, i.e. the acceleration of the hydrolysis of peptide substrates (Figure 2). In a separate native electrophoresis experiment, we also observed an increase in the brightness of the 26S proteasome band in the presence of 11S, but these data are not shown, since the mobility of 26S in the presence of 11S practically does not change.

The authors use a short peptide that presumably diffuse into open CP complexes and does not depend on ATPase activity and the poly glutamine stretches are very short compared to the physiological problematic poly glutamine stretches in Htt and likes.

Response: There is such a possibility; however, Table S1 shows that SDS significantly increases the activity of 20S, therefore, we assume that our 20S proteasome is in a closed state. Regarding the length of the glutamine peptide, this experiment cannot be performed with longer FRET substrates since the long distance between the donor and the acceptor will be a limiting factor for energy transfer. This reasoning has been added to the discussion. In addition, such oligopeptides are likely to be insoluble or poorly soluble (and prone to aggregation) in aqueous solutions (Chen and Wetzel, Protein Sci 2001), and it seems that such long polyglutamine sequences are soluble only when they are part of the native protein.

  • That both glutamine peptides are cleaved at smilar position does not convincingly show that cleave in middle of poly glutamine does not occur. There are too few controls and variations, the lysine and preferred binding of short peptide could influence and impact cleavage event that might be very different from stretch within larger poly peptide.

Response: Although LC/MS experiments have been performed repeatedly, we have never observed longer peptides containing Dabcyl. It is generally considered that there are three amino acid residues in the bonding zone in front of the hydrolysable bond (Huber et al. JACS 2015). Thus, if we saw the hydrolysis after KQ3, then all three positions would be occupied by glutamine residues, and if nothing interfered with hydrolysis anywhere, it would occur however we did not noticed hydrolysis near C-end of the Q10 peptide. This reasoning has been added to the discussion. Our data are also consistent with Venkatraman et al. Mol Cell 2004.

5 Complexes were purified from brain and it is described as brain related activity. However, are the purified complexes different from complexes purified from other organs? What data show this is unique from brain? This description seems more suggestive to provide perceived connection to neuronal disease than a have shown relevance for this highly in vitro study with artificial substrates.

Response: Constitutive proteasomes and immunoproteasomes usually coexist in cells, but the ratio between them depends on the cell type and the environment. Proteasomes from different tissues contain a different set of catalytic subunits, and normal brain proteasomes have been shown to consist mainly of constitutive subunits unlike the proteasomes from other tissues. The expression of proteasome regulators also varies between different cell types (Noda et al. Biochem Biophys Res Commun 2000, Turker et al. Cell Chem Biol 2021,  Davidson and Pickering, Front Cell Dev Biol 2023). All of the above data indicate the uniqueness of the proteasome from the brain.

Comments on the Quality of English Language

The message and intention of authors come across, however substantial editing would be desired to eliminate awkward English and remove unnecessary lengthy and indirect phrasing.

Response: Thank you for your comment, we proofread and revised the manuscript to improve the quality of English language.

Reviewer 3 Report

The 11S, also known as the PA28 regulatory complex, is an activator of the 20S proteasome, which is the catalytic core particle responsible for protein degradation. The 11S regulatory complex is reported to involve in non-ubiquitin-dependent protein degradation (PMID: 17512401). Furthermore, the study reported that 19S is not always attached to the 20S, and additional factors like (PA28/11S, PA200/BLM10, and ECM29) are also known to regulate the 20S complex activity. Importantly, the 11S regulatory complex is composed of three homologous subunits: REG alpha (also known as PA28α), REG beta (also known as PA28β), and REG gamma (also known as PA28γ). Each of these subunits can form a homoheptameric ring structure, and they can also form hybrid complexes with mixed subunits. When the 11S regulatory complex associates with the 20S core particle of the proteasome, it forms the 20S-11S complex or 20S-PA28 complex. An earlier study reported that up-regulation of 11S proteasome significantly enhanced proteasome-mediated removal of misfolded proteins (PMID: 21098724). Moreover, 11S binding allows the 20S core particle’s gate to open by binding to the interface of α-subunits (PMID: 11081519).

Polyglutamine (poly Q) is a stretch of consecutive glutamine residues in a protein, and when this region is expanded beyond a certain threshold, it can lead to the formation of protein aggregates, which are characteristic of several neurodegenerative diseases, such as Huntington's disease. The exact mechanisms by which the 11S complex recognizes and degrades polyglutamine-containing proteins are not fully understood. Here in this current manuscript, the heavy lifting here is done through in vitro experiments, kinetic assays and mass-spec approach, and docking study, which overall provided clarity to these complex processes linked with polyQ degradation.  The authors here developed novel model substrates to test the hypothesis and confirm the central aspects of the idea. For the first time, authors provided additional clarity on how the 20S and 26S proteasomes activity is stimulated by 11S, without altering the cleavage specificity.

Suggestions:

Overall, the manuscript is well written and accessible, and didn't find major flaws in the arguments or experimental design.  However, I would suggest the authors to improve the quality of the blots with a protein ladder (Figure S2 and Figure S3), and the electrophoresis gel (Figure S4). Additionally, the chromatographic profile (marking different proteasome complexes present) of the Isolated proteasome from lysate would be great.

Author Response

Suggestions:

Overall, the manuscript is well written and accessible, and didn't find major flaws in the arguments or experimental design.  However, I would suggest the authors to improve the quality of the blots with a protein ladder (Figure S2 and Figure S3), and the electrophoresis gel (Figure S4).

Response: Thank you for your suggestions, we improved these figures. Figure S3 contains no protein ladder as we use dot-blotting technique in this case. Figure S4 (Native electrophoresis) contains only one FITC-labeled protein, thyroglobulin (~700 kDa, right lane) as a molecular weight marker.

Additionally, the chromatographic profile (marking different proteasome complexes present) of the Isolated proteasome from lysate would be great.

Response: Thank you for the suggestion. We added chromatographic profiles to the supplemental file, Figure S6. In principle, one can refer to https://pubmed.ncbi.nlm.nih.gov/19496324/ and also reference 29 in the article where this protocol is described in more detail.